# The Use of 360-Degree Video to Reduce Anxiety and Increase Confidence in Mental Health Nursing Students: A Mixed Methods Preliminary Study

**DOI:** 10.3390/nursrep15050157

**Published:** 2025-04-30

**Authors:** Caroline Laker, Pamela Knight-Davidson, Andrew McVicar

**Affiliations:** 1Department of Nursing, School of Health & Medical Sciences, Clerkenwell Campus, City St George’s, University of London, London EC1V 0HB, UK; 2School of Nursing, Faculty of Health, Medicine and Social, Chelmsford Campus, Anglia Ruskin University, Chelmsford CM1 1SQ, UK; pamela.knight-davidson@aru.ac.uk (P.K.-D.); andy.mcvicar@aru.ac.uk (A.M.)

**Keywords:** 360-degree video, nurse education, stress and coping, mental health nurses, emotional distress, cognitive reappraisal, solution-focused brief therapy, VERA

## Abstract

**Background:** Stress affects 45% of NHS staff. More research is needed to explore how to develop resilient mental health nurses who face multiple workplace stressors, including interacting with distressed clients. Higher Education Institutions are uniquely placed to introduce coping skills that help reduce anxiety and increase confidence for pre-registration nurses entering placements for the first time. **Methods:** A convenience sample of first year mental health student nurses (whole cohort), recruited before their first clinical placement, were invited to participate. Following a mixed methods design, we developed a 360-degree virtual reality (VR) video, depicting a distressed service user across three scenes, filmed in a real-life decommissioned in-patient ward. Participants followed the service user through the scenes, as though in real life. We used the video alongside a cognitive reappraisal/solution-focused/VERA worksheet and supportive clinical supervision technique to explore students’ experiences of VR as an educative tool and to help build emotional coping skills. **Results**: N = 21 mental health student nurses were recruited to the study. Behavioural responses to the distressed patient scenario were varied. Students that had prior experience in health work were more likely to feel detached from the distress of the service user. Although for some students VR provided a meaningful learning experience in developing emotional awareness, other students felt more like a ‘fly on the wall’ than an active participant. Empathetic and compassionate responses were strongest in those who perceived a strong immersive effect. Overall, the supportive supervision appeared to decrease the anxiety of the small sample involved, but confidence was not affected. **Conclusion:** The use of 360-degree VR technology as an educative, classroom-based tool to moderate anxiety and build confidence in pre-placement mental health nursing students was partially supported by this study. The effectiveness of such technology appeared to be dependent on the degree to which ‘immersion’ and a sense of presence were experienced by students. Our cognitive reappraisal intervention proved useful in reducing anxiety caused by ‘the patient in distress scenario’ but only for students who achieved a deep immersive effect. Students with prior exposure to distressing events (in their personal lives and in clinical settings) might have developed other coping mechanisms (e.g., detachment). These findings support the idea that ‘presence’ is a subjective VR experience and can vary among users.

## 1. Introduction

Although work-related stress has pervasive negative effects across the NHS [1], it is essential to acknowledge the impacts for nurses, who are the largest staff group, and who have an integral role in the delivery of good quality care for large numbers of patients, daily [2]. Developing strategies that increase the ability to adapt and cope with significant sources of stress [3] may help nurses to better handle their stressful working environments.

In mental health nursing, the development of dispositional, psychological, and social attributes, which are fundamental in forming therapeutic working relationships, is key. This should begin in pre-registration nurse education and may be particularly helpful for student nurses prior to entering an acute practice placement for the first time. Empowering students to cope with stress by offering a platform to explore how they respond to stressful working situations may help students feel more in control [4] and support the development of coping skills. Universities might take a more active role in developing the emotional coping skills of student nurses by providing educational exposure to psychosocial challenges and potential confrontation in a safe and realistic pseudo-clinical environment.

As an educational tool, an immersive 360-degree video can be used to provide such an opportunity to help students explore how they manage stress and anxiety when placed in a stressful situation. Such scenarios can be mediated by virtual reality (VR), described as ‘a technological interface that allows users to experience computer generated environments’ in a safe and controlled manner [5]. VR technologies aim to parallel reality by creating immersive and interactive virtual worlds. A full experience is achieved when users believe that the scenarios they are experiencing actually simulate a real-world scenario [6]. Through such immersion, immediate therapeutic effects, such as improvement in mood [5,7] and reduction in psychological stress [8,9], have been achieved. Using experiential learning to acquire cognitive behaviour therapy skills has already proven successful amongst student nurses [10]. Therefore, in this study, an approach that combines 360-degree virtual reality (VR) video technology with a cognitive reappraisal strategy was explored.

### 1.1. Cognitive Reappraisal

Cognitive reappraisal is an evidence-based therapeutic approach that draws from cognitive behavioural therapy (CBT) [11]. It aims to explore the antecedents (thoughts and feelings) to an individual’s behavioural responses [12]. Through re-framing the meaning of the situation, it supports the modification of negative or challenging thoughts and emotions to reduce the emotional impact [13,14]. This enables more positive thoughts, emotions, and behavioural responses [11,15]. There are a number of studies which show a more positive adjustment to stress if an individual is able to reappraise their emotional responses, which can also positively influence psychological health [16,17,18]. Gross and John [19] have shown greater experiences and expressions of positive emotions in individuals who reappraise, using both self and peer-rated measures, which indicates that these strategies, once acquired, can be sustained.

Therefore, teaching student nurses to use a cognitive reappraisal technique might usefully provide a framework that increases their sense of being in control in stressful situations, to reduce anxiety around handling conflict and working with emotional distress, events which are part of daily life in mental health wards.

### 1.2. VERA and Solution-Focused Strategies

In addition, solution-focused brief therapy principles [20,21,22,23,24,25] and the VERA cycle for communication (V = validation, E = emotion, R = reassure, and A = activity) have shown promise in promoting confidence amongst staff working with distressed individuals [26,27]. VERA has been used successfully to work with people with dementia and psychosis [26,27].

### 1.3. Aims

The overall aim of this study was to examine whether using a 360-degree video scenario was helpful in preparing mental health nursing students to cope with the potentially stressful experience of working therapeutically with a highly distressed patient, by supporting them to think through their responses.

The research questions (RQ) were as follows:○RQ1: What was the impact of VR as an educational tool and learning experience?○RQ2: How did student mental health nurses perceive their abilities to develop a therapeutic nurse/client relationship in a stressful clinical situation?○RQ3: Did working through a process of cognitive reappraisal and clinical supervision help students to envisage how they might cope better with a stressful interaction, in terms of increased confidence and reduced anxiety?

### 1.4. Study Design

This study used mixed methods, by combining the use of multi-media, qualitative inquiry, and scaling as part of a supportive clinical supervision approach, using a cognitive reappraisal/solution-focused/VERA worksheet.

## 2. Materials and Methods

Sample: Participants in a convenience sample from an entire cohort (n = 21) of first year student nurses, enrolled in a BSc mental health nursing course, were approached and invited to participate before they went into their first placement. The student nurses were presented with participant information and consent sheets and offered an opportunity to discuss the study with a member of the research team.

### 2.1. Methods

A protocol paper for this study has already been published [28], which details this study and includes the materials and methods used. In brief, all participants were invited to watch the video, and all participants undertook a one-to-one reflective discussion with a researcher afterwards. However, a subgroup of randomly selected participants also engaged in the supportive clinical supervision approach, using the cognitive reappraisal/solution-focused/VERA worksheet. In each case, the whole process was video-recorded; hence, the primary method of data collection was qualitative, as data were captured via video interviews of the participants as they moved through the process of engagement with the 360-degree video and reflection (and the clinical supervision intervention, for those in the subgroup). However, as rating scales were used to examine changes in confidence and anxiety as a result of engaging in the clinical supervision intervention, it can be considered that this study uses mixed methods.

### 2.2. Materials

Ahead of participant recruitment, some preparatory work was required for this study. A realistic video scenario was filmed in a decommissioned mental health ward, using 360-degree technology (scene capture from every direction) and high levels of sensory fidelity (visual and auditory cues) to replicate a real-life experience. The aim was to allow students to experience the scenes as though they were there by viewing the video through a VR headset. The film was approximately 5 min in length. Actors played the role of service users and nurses. A service user called ‘Mary’ was the central character. Service user involvement was sought during the creation of the video to ensure sensitivity, appropriateness, and well-roundedness [29].

There were three scenes:
Scene 1: In a corridor, Mary is beginning to show signs of stress, asking to go home, and in frustration, she throws tea from a cup she is holding towards the viewer.Scene 2: Mary is seen crying in her bedroom and is clearly very distressed.Scene 3: Mary is showing frustration whilst interacting with two men in the dayroom.

Questions to enable participant reflection after watching the video were developed through consultation by the research team, which included two mental health nurses, one adult nurse, and a professor of stress responses in clinical or workplaces.

In addition, a cognitive reappraisal/solution-focused/VERA worksheet was designed by modifying a pre-existing tool [30] through consultation with a systemic therapist who was a member of the research team, to create a framework for students to work through, supported by a clinical supervisor to consider how to respond to the distressed service user portrayed in the 360-degree scenario. The worksheet guided participants to explore and rate their emotional response to and thoughts/beliefs about the video, with a particular focus on anxiety and confidence. The final section took participants through solution-focused/VERA-based questions, which aimed to boost confidence and reduce anxiety by focusing on using pre-existing skills and abilities to support the distressed service user. The items are outlined in full in a protocol paper [28].

### 2.3. Process

The process for participating in this study and for collecting data was as follows:

All participants (n = 21) watched the 360-degree scenario, individually, through a VR headset. Research questions 1 and 2 were explored using data from this group.

○RQ1: What was the impact of VR as an educational tool and learning experience?○RQ2: How did student mental health nurses perceive their abilities to develop a therapeutic nurse/client relationship in a stressful clinical situation?

After watching the 360-degree scenario, all participants were asked to reflect on the experience. The questions devised to support these reflections are outlined in a protocol paper [28]; however, some examples can be seen below:○In the video, the service user is talking to someone. In your view, who was that person?○How did you feel whilst watching the scenario?○How able were you to relate to the service user?○How did you feel overall about the 360-degree video experience?

A clinical supervision subgroup (n = 11) also received a supportive clinical supervision session within a cognitive reappraisal/solution-focused/VERA framework, which followed a modified worksheet [28]. Research question 3 was explored using data from this subgroup.

○RQ3: Did working through a process of cognitive reappraisal and clinical supervision help students to envisage how they might cope better with a stressful interaction, in terms of increased confidence and reduced anxiety?

After the first viewing, students were instructed to use a scale of 0 (not intense at all) to 10 (the most intense ever felt) to address the following:○Identify emotions: as part of the process of cognitive reappraisal, each student was asked to describe their feelings after watching the scenario and rate the intensity of their emotional responses.○Evaluate thoughts and rate how strongly they held these beliefs.○Rate their levels of confidence and anxiety.

Participants were also instructed to explore how they might respond in the situation, using solution-focused brief therapy skills and by drawing from the VERA cycle. The full details are available in a prior publication [28]; however, an example is as follows:

Think about a situation in your past where you dealt successfully with an anxiety provoking incident.

○What did you do then to successfully manage the situation? ○What did you say, keep in mind, feel, or remember that helped you solve the situation in a positive way?

### 2.4. Data Analyses

The data analysis strategy was underpinned by social phenomenological theory [31], focusing on the interaction between participants (students) and a distressed client (Mary) and the students’ understandings, sense-making, and constructed meanings, derived from the virtual world around them; how the students interpreted and responded to the stressful clinical situations they were in; and whether interacting in a more experience-based clinical setting, using a structured clinical supervision session drawing form cognitive reappraisal, SFBT, and the VERA framework, helped students to cope better with this stressful interaction (in terms of reduced anxiety and increased confidence). Audio–visual data were collected to allow an exploration of verbal and non-verbal content to further inform our understandings and interpretations of the interactions between students and ‘Mary’ in the virtual world. All data were transcribed and as the data used audio and visual formats, a thematic content analysis was conducted to allow flexibility and to allow a deeper consideration of social interaction [32,33]. Nvivo 11 software was used to code the data.

Through line-by-line discussions of the transcribed data, alongside the audio–visual data, an inductive approach was taken to generating themes within the coding frame. The different backgrounds of the team, which comprised two nurse researchers with backgrounds in mental health (CL) and adult nursing (PK-D), and a researcher with a background in physiology and healthcare (AMcV), informed the discussion to ensure a deep engagement with the data and a reflexive approach to constructing shared meaning [34,35].

The rating scales collected from the subgroup participants, before and after the clinical supervision intervention, were analysed using Stata 15 to plot graphs, based on paired t-tests. However, it is important to note that given the size of the sample (N = 11), these analyses were entirely exploratory and only used to make sense of the data, which were considered in relation to the qualitative interviews, to augment our understanding of the emotions reported by the participants.

## 3. Results

Sample: There were N = 21 participants in total, of which 11 participated in the clinical supervision, as indicated in Table 1.

### 3.1. Summary of Qualitative Findings

Our analysis of the reflections of the whole group revealed that the behavioural and verbal responses to the distressed patient scenario were varied. Observationally, participants also appeared to display different immersive effects in terms of their engagement with the 360-degree video content. Some students reported that they had used VR technology, which may have been a factor in how they responded. Overall, the participants responded positively to the experience and enjoyed the VR scenario. All students agreed that it was a valuable learning experience.

We observed four prominent, qualitative themes, that indicated an interplay between emotion, agency, and presence, which are presented with data below:Theme 1: strong presence effects/high levels of empathy;Theme 2: voyeuristic stance/strong emotional response but restricted agency;Theme 3: detached from distress/reduced emotions and agency;Theme 4: perceptions of risk.

These themes were associated with the degree to which the students wanted to help Mary. This was either observed via behaviours that manifested in the room, which suggested that they wanted to help (e.g., reaching out to her), or through their verbal reports of the potential they had to help.

Theme 1: strong presence effects/high levels of empathy

In these data, the experience was reported by students as highly immersive and emotionally evocative. Qualitatively, some participants reported strong, presence effects of ‘almost being there’, and these participants demonstrated more empathetic responses to the distress of the service user. Indeed, one student felt the experience so intensely that they themselves became distressed, and we had to stop the video.

Participant 12 claimed the following:‘I feel kind of sad that she was being neglected. That is person is supposed to be looking after me and is ignoring me. That is disrespectful, so I can’t get the best from that place. So, it makes me feel unhappy putting myself in that position…’

Participant 11 claimed the following:‘It’s like she weren’t being heard so even though she was directing it all at me I could… I felt the same at the same time’.

Theme 2: voyeuristic stance/strong emotional response but restricted agency

Some students expressed a perceived voyeuristic stance, where they felt ‘like a fly on the wall’. These students felt emotionally connected to the scenario, but they were frustrated by their perceived lack of agency as they were unable to interact with Mary. Where a voyeuristic stance was observed, the participants reported that inability to interact led to disempowerment, frustration, and sadness. This was in part due to the VR preventing interaction with the service user.

Participant 1 claimed the following:‘I wanted to intervene, but I wasn’t really sure. The woman was distressed, and I think she needed to talk to someone, or somebody needed to speak to her, but I couldn’t really communicate, and when I did there was no response back— one way traffic. Initially it was like…if I was to use to word strange or shocked, I couldn’t really…it was, Oh, what do I do? What do I say? What’s going on?’.

Participant 5 claimed the following:‘I still feel guilty, if I was, if that was a scenario like in real life, you know, I wouldn’t go over and see, I wouldn’t…Yes, it’s probably now I’m actually getting things out, it’s actually making me really disappointed’.

Theme 3: detached from distress/reduced emotions and agency

Some students appeared to be detached and immune to the distress of the service user, displaying a reduced emotional response and less of a desire to act to support Mary. Upon viewing the videos, outwardly they showed no emotion or intention to act. After discussions, this appeared to be linked to previous exposure to client distress.

The researcher asked the following question:‘So, how intense were your feelings during the experience?’

Participant 10 replied as follows:‘Probably about a 5 maybe. I think because I’ve done it before, unfortunately I’ve probably seen it. So, to someone that hasn’t seen it, it would be more upsetting, wouldn’t it? Not that I’ve been desensitised to it, but yeah’.

Theme 4: perceptions of risk

There were interesting variations in how individuals interpreted the scenario and its themes, which seemed to relate to how much exposure students may have previously had to the clinical setting. For those with prior experience, issues of perceived risk, either to Mary or to themselves, were a greater topic of concern than their own compassionate response.

Participant 10 claimed the following:‘What might happen? Hurt herself…hurt somebody else. She was trying to break through a door so she might have broken through the door or smashed a window…or upset one of the men that were in the room because, by the sounds of it, they weren’t very well either so she could have got hurt.’

Participant 2 claimed the following:‘Actually, when I was looking there and I saw the two guys, I was a bit distressed because I was thinking one is going to attack the other. So I was anticipating something bad might happen so that was distressing me more. And the last being distressed wasn’t as distressing as compared to when I was anticipating bad’.

### 3.2. Clinical Supervision: Subgroup: Self-Reported Emotions and Thoughts and Ratings of Confidence and Anxiety Pre and Post Clinical Supervision

In the clinical supervision subgroup, we explored the participants’ emotional responses to the distressed service users, with a particular focus on confidence and anxiety.

This was accomplished by examining whether having a supportive clinical supervision session, with cognitive reappraisal/solution-focused/VERA questions designed to help participants focus pre-existing skills and abilities, helped improve confidence and reduce anxiety.

Anxiety ratings by students are shown in Table 2 below:

In fact, self-rated perceptions of anxiety varied widely across the subgroup, as shown in Figure 1.

Self-rated anxiety ranged between three and ten during the viewing. There was little shift, overall, in how anxious the students felt as a result of the supervision; the self-rating of anxiety ranged from two to ten. Only one student had notable shift, participant 8, who reported a six-point drop in anxiety level after the supervision. Less noticeable shifts were reported by participants 9 and 12. Students explained higher anxiety scores alongside feelings of discomfort, helplessness, or guilt in the absence of being able to offer support to Mary.

Participant 5 claimed the following:‘Yes, I was feeling anxious, because you know I couldn’t do anything. And if I could probably that would have reduced the anxiety and everything else. And probably at the beginning of it I just didn’t’ know what to expect, and I was just looking at these walls and these doors around me, I just didn’t know what was going to come out or what was going to happen. So that was a bit intense. It’s for Mary yeah, but especially when she was in the room, and she was really upset. And nobody bothered to actually go in there and find out what was going on. Why she was so distressed and upset. And the other thing is I feel really angry because the door she couldn’t get out of—that room—if she wanted to go for a walk or if she went to try to open the door, and the door was locked.

There was also some anxiety about being in the virtual world.

The researcher asked the following question:‘How anxious did you feel when viewing this situation on a scale of 0–10?’

Participant 1 replied as follows:‘I was very anxious because the first thing is, “Okay, who do I talk to? Who can help me?” I looked around. Then that’s when I asked, “Can you talk? Can you talk to us?’

In line with the detachment or immunity to distress that was described earlier, lower anxiety scores were founded on having had previous exposure to similar clinical situations.

Participant 6 claimed the following:‘To be honest I didn’t feel anxious, because I’ve seen worse. However, I think this lady in particular, she’s not causing a problem, so because I’ve seen worse, so I wasn’t anxious or anything. I just feel that maybe she was; maybe she was being neglected. She’s not young as well, I don’t know, because its very difficult you know. If she is always someone that’s seeking attention who [inaudible] or whatever she would do something, maybe I’d understand better and think, “Okay, this is the reason”. Maybe she would hurst somebody or slap somebody or whatever, but I don’t know that.”

We also explored how confident students felt to offer support to Mary during the 360-degree video scenario and whether having a supportive clinical supervision session helped improve their confidence, as shown in Table 3 below:

Most students felt very confident that they could offer support to Mary to help her move on from her distress, as shown in Figure 2.

Confidence ratings were generally high (between six and ten), with only one participant indicating low confidence (participant 6, whose confidence rating was four) during the viewing. For most participants, the confidence level remained the same or increased after the supervision. There was one exception, participant 3, who reported lower confidence after the supervision.

## 4. Discussion

We set out to explore whether we could use a 360-degree virtual reality scenario of a distressed patient as an educative tool in a group of student nurses ahead of their first placements. The aim was to give them an opportunity to reflect on their own thoughts, emotional responses, and behaviours to the distress shown by the service user in the scenario. We then supported the students using a clinical supervision intervention to reappraise their thoughts and emotions and consider what appropriate responses they might offer to the distressed service user.

The VR technology gave us a medium to replicate a number of elements of real-world mental health nursing practice, such as the locked ward environment, a sense of feeling unheard, the emotional distress of a service user, and a wider social interaction between staff and other clients.

### 4.1. What Was the Impact of VR as an Educative Tool and Learning Experience?

Worthy of discussion was the finding that the VR technology and scenario inducted a sense of presence, as distinct from immersion, which is an individual and context-dependent user response related to the experience of being there [36], and is usually understood in terms of the ability of the technology to evoke the real world. Given the 360-degree video was developed with a service user with lived experience of mental illness, it is likely that its authenticity was enhanced due to her contribution [29]. Presence is assumed to have many beneficial effects, including improvements to learning and understanding, and is a useful adjunct in education that has significant practice-based elements, such as nursing [37]; however, as was the case in this study, users can experience different levels of presence with the same system at different times, depending on state of mind, recent history, and other factors. If presence impacts learning, then it may be supposed that the impact of VR varies, depending on the level of presence achieved.

In mental health nursing, the ability to interact socially and to understand yourself in relation to the wider social world is fundamental to developing therapeutic relationships. This is an area of education that is currently lacking in nursing curricula, and although emotional awareness and skills for coping are addressed through clinical practice learning, and supported through practice supervision, there have been longstanding concerns, now exacerbated by the effects of the pandemic, that if mental health nursing staff do not receive the support they need [38], there is a likely knock-on effect for student nurses, who therefore need focused support to develop coping strategies.

### 4.2. How Did Student Mental Health Nurses Perceive Their Abilities to Develop a Nurse/Client Relationship in a Stressful Clinical Situation?

Current VR research shows that emotion and presence are correlated [37], following the laws of emotion whereby the more real a person perceives a situation to be, the higher the intensity of their emotional response [39]. Therefore, in this study, we were also interested in additional presence concepts, such as self-presence, or how much of the virtual self within the virtual environment was psychologically regarded by users as their actual self, and social-presence, or how much emotional inter-connectedness is created though the virtual environment [37]. Evidence of both social and self-presence effects would indicate that the 360-degree VR scenario allowed for the generation of authentic emotional responses, providing some evidence that this platform can be used in mental health nursing to develop training around emotions and emotional responses.

In this study, we were therefore interested in how the students understood distress, empathised, and developed an appropriate, compassionate response. Empathy is notably the antecedent emotion to compassionate action, if we understand compassion to be a sensitivity to distress that prompts action to alleviate suffering [40]. We observed a spectrum of effects in the student’s responses, as shown in Figure 3:

### 4.3. Strong Presence Effects/High Levels of Empathy

In some students, we observed strongly empathetic emotional responses to the service user’s distress, which motivated them to act to help. For example, some students showed a profound sense of empathy as a vicarious response to the service user’s emotions [41], beyond our expectations. However, it was clear that the depth of their compassion also left them carrying some difficult and conflicting emotions if they wanted to help but were not quite sure of what to say or do to help, which left students with some unresolved intense feelings after the interaction. Clearly, more consideration is needed around the ethics of using immersive experiences in the classroom, which can trigger strong emotional responses from students, and it will be important to ensure that appropriate and sufficient support is offered in the form of supervision and debriefing when dealing with topics around emotional distress.

### 4.4. Voyeuristic Stance/Strong Emotional Response but Restricted Agency

In other students, their strongly empathetic response in relation to the distress observed in the service user was impaired by the VR technology, which did not allow for a two-way interaction; hence, they were unable to enact a compassionate response, which produced feelings of paralysis, powerlessness, and inaction in the students, and in some cases, caused distress. This is of interest, as we have seen evidence of feelings of powerlessness leading to inaction and avoidance in previous work [42].

### 4.5. Detached from Distress/Reduced Emotions and Agency

Some students maintained an emotionally detached stance from the distressed service user, citing previous exposure to similar experiences. Whether these findings give us some insight into the early development of burnout would be worthy of further consideration, as similar emotional responses are visible in research studies looking at the care provided by nurses in real-world acute mental health services, where we see evidence of burnout and de-motivation in the form of defensive or avoidant practice, where staff unconsciously or consciously protect themselves from over exposure to emotional distress [42,43,44]. Future research might usefully consider whether burnout is an issue within mental health nursing cohorts.

### 4.6. A Need for Increased Focus on Reducing Anxiety and Promoting Confidence, or a Wider Mental Health Concern?

More generally, this study spotlights a major concern relating to the mental wellbeing of student nurses. There is increasing evidence of poor mental health amongst the nursing workforce in the UK [45]. In nursing courses in the UK, USA, and around the globe, attrition is high, with many student nurses struggling with depression, anxiety, distress, or burnout [46]. An unforeseen effect of the VR experience was its ability to highlight those students who had become detached from the emotions displayed by Mary, which, interestingly, may be a precursor to burnout, which is a well-known phenomenon in mental health work. The findings of this study highlight the need for nurse educators to embed learning that enables coping skills. Often, curricula have a superficial focus in this area, where clearly, more in depth focus is required. From a classroom perspective, given the fact that the VR technology and scenario was able to produce strong presence effects among some students, this makes it an effective tool where real-life simulation would be beneficial, but not realistic to achieve. As not all students might experience such an effect and predicting/anticipating which students might do so is not possible, the utility of this VR experience alongside a reappraisal intervention in planned curricula is challenging. We did however find that the experience offered a resource for students to understand their responses to the scenario in relation to the empathetic and compassionate curriculum they were undertaking. From this standpoint, the VR experience might be beneficial for all students who experience it, whether or not a strong immersive effect is achieved.

### 4.7. Did Working Through a Process of Cognitive Reappraisal and Clinical Supervision Help Students to Imagine How to Cope Better with a Stressful Interaction, in Terms of Increased Confidence and Reduced Anxiety?

In evaluating whether the clinical supervision intervention was helpful, there was little change in student confidence as a result of participation, as these scores suggested that students were already very confident that they could help Mary to reduce her distress. Although this is encouraging, it may also reflect their lack of experience in understanding how to work therapeutically with people in distress and the range of communication skills and depth of resilience needed to work with someone who does not want to remain on a ward. There was a little more of a shift in how students rated their anxiety as a result of the supervision, with slight improvements shown.

However, students verbally expressed their views of how participation in the VR experience had affected them and the analyses produced some interesting insights into how students interpreted and responded to their own emotions, and those of the service user in the scenario. This approach might provide a useful platform to build on here, as the whole process gave students a supportive opportunity to reflect on and engage with how they felt when placed in a stressful situation, as well as positively highlighting the skills they already have and that they can harness to support a person in distress, which is a much needed and often lacking element of learning in our current mental health curricula.

### 4.8. Limitations

We recognise that there were limitations with the scenario, as the students had limited agency to interact with the service user; however, the emotions that were generated were relatable to real-world practice and this approach may therefore be a useful way to engage mental health nursing students with concepts around their own ‘self’ responses, how to access support, and work on self-care to promote emotional coping skills. Furthermore, the sample size in this study was small, drawn from a convenience group of student nurses from one University setting only, and qualitative in nature. Although some fascinating themes arose from the data collected, more research is clearly needed before any conclusions might be drawn.

## 5. Conclusions

Further research is needed to reduce anxiety and build confidence in mental health student nurses, who are subjected from early in their training to clinical settings with multiple stressors, including interacting with an acutely unwell and distressed client group. In this study, we used a 360-degree virtual reality (VR) scenario alongside a cognitive reappraisal strategy/supportive clinical supervision technique to help build emotional coping skills in our mental health student nurses.

Although there was little shift in students’ perceptions of confidence and anxiety as a result of the intervention, the qualitative analysis suggested that the VR scenario was an effective was to engage and support students to learn about distress and how to cope in stressful situations. Therefore, Higher Education Institutions should consider using VR and supportive clinical supervision to increase emotional self-awareness and to build coping skills in pre-registration mental health nurses, particularly before students enter placements for the first time.

## Figures and Tables

**Figure 1 nursrep-15-00157-f001:**
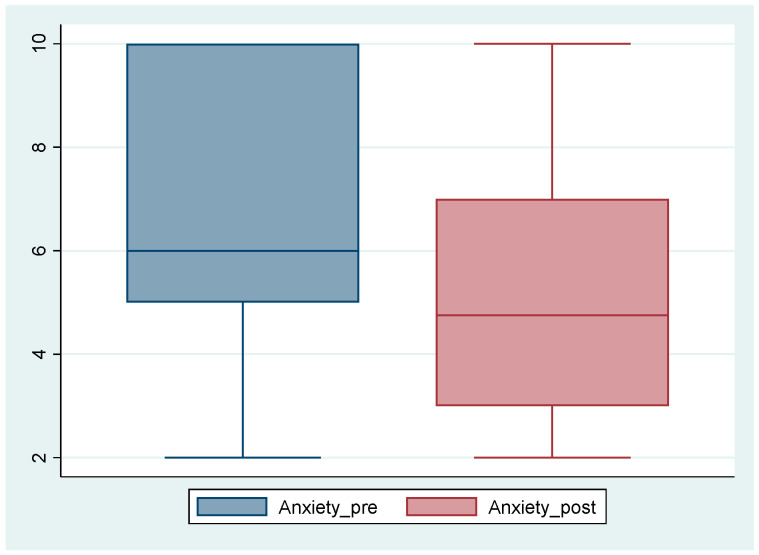
Anxiety scores pre/post clinical supervision.

**Figure 2 nursrep-15-00157-f002:**
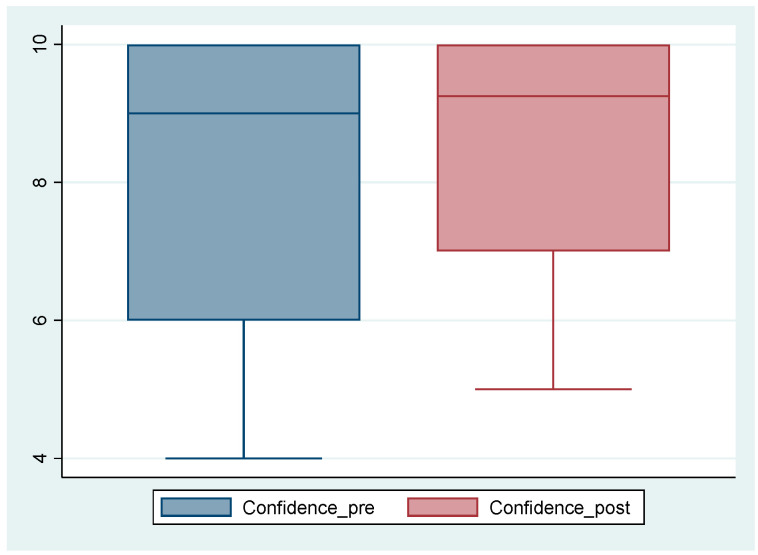
Confidence scores pre/post clinical supervision.

**Figure 3 nursrep-15-00157-f003:**
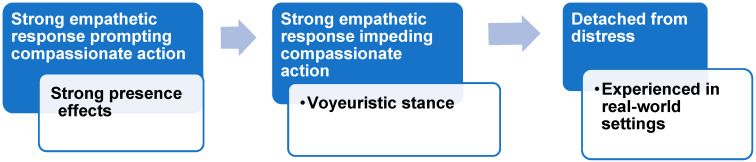
Spectrum of effects.

**Table 1 nursrep-15-00157-t001:** Demographic data.

Sample Data		Whole Group	Subgroup
N ^1^		21	11
Gender	Male	9	4
	Female	12	7
Age	Ranges from	21–63	30–56
Ethnicity	Black African	14	9
	White British	5	2
	Black British	1	0

^1^ 21 students in total participated in this study and within that sample, there was a subgroup of 11 students who also participated in the supervision.

**Table 2 nursrep-15-00157-t002:** Anxiety: pre and post clinical supervision.

Participants (N = 11)	How Anxious Were You on a Scale of 1–10?	
	During the 360-Degree Video	After the Clinical Supervision
1	10	9
2	7	Missing data
3	2	3
4	6	6
5	10	10
6	3	2
7	5	4.5
8	9	3
9	6	4
10	5	5
11	10	7

**Table 3 nursrep-15-00157-t003:** Confidence: pre and post clinical supervision.

Participants (Intervention Group N = 11)	How Confident Were You on a Scale of 1–10?	
	During the 360-Degree Video	After the Clinical Supervision Intervention
1	10	10
2	6	No data
3	10	7
4	6	7
5	10	10
6	4	5
7	9.5	9.5
8	8	9
9	9	10
10	8	8
11	10	10

## Data Availability

Data is contained within the article.

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
