# Peer review of "The Use of 360-Degree Video to Reduce Anxiety and Increase Confidence in Mental Health Nursing Students: A Mixed Methods Preliminary Study"

_nursrep, 2025, doi:10.3390/nursrep15050157_

Round 1

Reviewer 1 Report

Comments and Suggestions for Authors

The title is misleading - it is more about the experience than the further development of coping strategies
There should be no brackets in the title
The abstract should explain the situation depicted in the VR simulation
The introduction should not begin with the protocol. This should be mentioned additionally in the method.
The relevance of VR should be emphasised before the research question (line 57)
The research question should be revised again. Is it really about resilience, or the experience of anxiety and self-confidence?
This should be explained before VERA is mentioned (line 87)
All instruments must be explained in detail in Material and Methods
Isn't the study more of a mixed methods design?
How are the different data triangulated?
How is resilience analysed?
The result categories must be described in the methodological procedure
The interview results must be summarised (line 215 ff.)
How are the different data collections (qualitative and quantitative) combined?
Results are localised in the discussion (line 389)

Author Response

Dear Reviewer 1

Many thanks for your comments. We have attended to the issues flagged as follows:

Reviewer 1

Response

Details

The title is misleading - it is more about the experience than the further development of coping strategies

Thank you we have revised the title. NB reviewer 2 suggested he following change to the title: The use of 360-degree video to reduce anxiety and increase confidence in mental health nursing students

Title: Mental health Nursing students experience of a 360-degree video to support emotional coping skills

Changed to [lines 1-3]:

Title: The use of 360-degree video to reduce anxiety and increase confidence in mental health nursing students

There should be no brackets in the title

Thank you. Removed.

Lines 1-3

The abstract should explain the situation depicted in the VR simulation

Thank you, this is a valid observation, and we have taken the opportunity to make changes to the Methods section in the abstract.

Lines 12-16

The introduction should not begin with the protocol. This should be mentioned additionally in the method.

Thank you. This has been removed and now appears in the method section.

Please see line 116

The relevance of VR should be emphasised before the research question (line 57)

Thank you for flagging this. We wonder if including sub-headings which were phrased as questions, was mis-leading. It appears that perhaps these may have been appeared to be research questions. We apologise if this was not clear and hope that these changes have addressed the issue raised. The research questions can be seen in the in the Aims section. These have been labelled more clearly.  

Please see lines 97-109. 

The research question should be revised again. Is it really about resilience, or the experience of anxiety and self-confidence?

We have removed the sub-headings which were phrased as questions, which may have been confusing to the reader. The research questions have been labelled more clearly and can be seen in Aims section.

Please see lines 97-109

This should be explained before VERA is mentioned (line 87)

Thank you for your comment and apologies, as it was not clear exactly what was meant here – however, we have deleted line 87, which was perhaps superfluous.

Deleted line 87 in original manuscript.

All instruments must be explained in detail in Material and Methods

Thank you for your comment. The methods section has been completely re-written to make it clearer.

Lines 110-201

Isn't the study more of a mixed methods design?

Thank you for flagging this. I can see what you mean, as there are self-reported rating scales included. However, these were completed as part of the process of interviewing each participant and the whole experience was captured via video recordings. The interactions were transcribed and analysed using qualitative methods, so this was primarily a qualitative study. However, the inclusion of the rating scales using graphs to compare the scores pre and post receiving the clinical supervision intervention does lend itself to a mixed methods design, and this has been noted in the methods section.

Please see lines 116-128

How are the different data triangulated?

Please see above. In addition, a section has been added to the Analysis section to explain that the analysis of the rating scales was analyses were entirely exploratory and a device only to make sense of the data, which were considered in relation to the qualitative interviews, to augment our understanding of the emotions reported by the participants.

Please see lines 224-229

How is resilience analysed?

We have not specifically analysed resilience in this study. We were to some extent considering that reducing anxiety and building confidence might be seen as part of resilience building. However, we accept that resilience can be understood as a stand-alone concept. Therefore, we have de-emphasised ‘resilience’ as a term in this paper, as the focus is more about how participants responded to the stressful scenario in terms of their perceptions of their own confidence and anxiety.  

This has been done throughout the paper.

The result categories must be described in the methodological procedure

The results are primarily qualitative and were produced via an inductive process of analysis, hence the themes are not articulated in the methods section.

However, we have re-written the methods sections to set out more clearly how the data were collected, which was via video recordings that captured how participants engaged with the 360-degree content, followed by a process of reflection. Examples of the reflection questions are provided. A sub-group also underwent a supportive clinical supervision process using a worksheet drawing from principles of cognitive reappraisal, solution focused therapy and VERA. Example topics and questions are provided.

Please see lines 110-201.

The interview results must be summarised (line 215 ff.)

Thank you, we have taken the opportunity to summarise the results more clearly.

Lines 236-253

How are the different data collections (qualitative and quantitative) combined?

Thank you. I think perhaps it was confusing in the first submission, because of an absence of information in the methods section, which has now been re-written to reflect that although there are self-reported rating scales included, these were completed as part of the process of interviewing each participant and the whole experience was captured via video recordings. The interactions were transcribed and analysed using qualitative methods, so this was primarily a qualitative study.

Paired t-tests were used to plot graphs as a visual aid to explore the rating scales data. However, it has been noted that these analyses were entirely exploratory due to the small sample size (N=11), and a device only to make sense of the data, which were considered in relation to the qualitative interviews, to augment our understanding of the emotions reported by the participants. 

The rating scale data are compared with the qualitative findings in the results section.

Please see lines 224-229.

Please see lines 324-387

Results are localised in the discussion (line 389)

Thank you we have removed this to the results section.

Alteration made to line 335.  

We hope that these amendments address the issues raised. Please do let us know if anything further is required. Many thanks for your kind review.

Reviewer 2 Report

Comments and Suggestions for Authors

The topic addressed in this study is highly relevant, and the results obtained from such research are essential for improving and optimising the training of healthcare professionals working in high-stress environments. However, although the study is explicitly exploratory and qualitative, I find the methodology and statistical analysis rather limited and not sufficiently robust, thus making it difficult to draw even preliminary conclusions on the subject.

Below are more specific comments:

  • Title: I recommend modifying the title to: The use of 360-degree video to reduce anxiety and increase confidence in mental health nursing students.
  • Abstract: It would be helpful to include more specific methodological details in the abstract (e.g., how variables were assessed, sample size, and significant results).
  • Page 2, Line 69: Given the extensive literature on VR in medical settings (both with patients and healthcare professionals), why reference military studies?
  • Line 74, Reference 12: Possibly a typographical error.
  • Line 89, Reference 17: Left bracket appears to be a typographical error.
  • Sample size: There is a total lack of sample size calculation. At the very least, indicate that it was a convenience sample, but this limitation should be appropriately acknowledged in the discussion and in the conclusions.
  • Page 5, Lines 197-202: These considerations would be more appropriate in the introduction and/or discussion. The results section should present only the findings, described in a neutral and descriptive manner.
  • Page 5, Lines 202-203: “We observed an interplay between emotion, agency, and presence in our data” – How was this interaction observed? No statistical details are provided. The entire paragraph is purely narrative, lacking any statistical results, including descriptive statistics.
  • Sense of presence: It is unclear how the sense of presence was assessed. There are various validated scales and ad hoc questionnaires available for this purpose.
  • Additional measures: Including a burnout questionnaire could be valuable to explore potential correlations with the results obtained. For instance, the Authors mention a detached response from some students; could this correspond to the cynicism factor of burnout?

In conclusion, while the study addresses a compelling topic, improving methodological rigour and providing more detailed analyses would significantly enhance the reliability and impact of the findings.

Author Response

Dear Reviewer 2

Many thanks for your comments. We have attended to the points raised and made amendments as follows:

Reviewer 2

Response

Details

Title: I recommend modifying the title to: The use of 360-degree video to reduce anxiety and increase confidence in mental health nursing students.

Thank you for your suggestion. This has been amended.

Lines 1-3

Abstract: It would be helpful to include more specific methodological details in the abstract (e.g., how variables were assessed, sample size, and significant results).

Thank you for your suggestion. This has been amended, noting that this study is primarily qualitative in design with some focus on self-reported rating scales

The abstract has been completely re-written to make it clearer

Lines 8-32

Page 2, Line 69: Given the extensive literature on VR in medical settings (both with patients and healthcare professionals), why reference military studies?

Thank you. We agree that this appears superfluous and has been deleted.

Deleted line 87 in original manuscript.

Line 74, Reference 12: Possibly a typographical error.

Thank you. Removed.

Line 63

Line 89, Reference 17: Left bracket appears to be a typographical error.

Thank you. Altered.

Lines 74

Sample size: There is a total lack of sample size calculation. At the very least, indicate that it was a convenience sample, but this limitation should be appropriately acknowledged in the discussion and in the conclusions.

Thank you. This was primarily a qualitative piece of work. I think perhaps it was confusing in the first submission, because of an absence of information in the methods section, which has now been re-written to reflect that although there are self-reported rating scales included, these were completed as part of the process of interviewing each participant and the whole experience was captured via video recordings. The interactions were transcribed and analysed using qualitative methods, so this was primarily a qualitative study.

Paired t-tests were used to plot graphs as a visual aid to explore the rating scales data. However, it has been noted that these analyses were entirely exploratory due to the small sample size (N=11), and a device only to make sense of the data, which were considered in relation to the qualitative interviews, to augment our understanding of the emotions reported by the participants. 

Thank you. This has been added.

Please see lines 110-126.  

Please see lines 224-229

Page 5, Lines 197-202: These considerations would be more appropriate in the introduction and/or discussion. The results section should present only the findings, described in a neutral and descriptive manner.

Thank you. We agree. This has been altered.

Please see lines 235-252

Page 5, Lines 202-203: “We observed an interplay between emotion, agency, and presence in our data” – How was this interaction observed? No statistical details are provided. The entire paragraph is purely narrative, lacking any statistical results, including descriptive statistics.

Thank you, this has been re-written to emphasise that these are qualitative findings.

Please see lines 243-244

Sense of presence: It is unclear how the sense of presence was assessed. There are various validated scales and ad hoc questionnaires available for this purpose.

Thank you, it has been clarified in the text that the findings were based on qualitative reports by participants.

Please see lines 255-258.

Additional measures: Including a burnout questionnaire could be valuable to explore potential correlations with the results obtained. For instance, the Authors mention a detached response from some students; could this correspond to the cynicism factor of burnout?

Thank you. We are of the same view that future research could consider how burnout features in nursing student cohorts.  

Added a sentence: lines 468-9

In conclusion, while the study addresses a compelling topic, improving methodological rigour and providing more detailed analyses would significantly enhance the reliability and impact of the findings.

We hope that these amendments address the issues raised. Please do let us know if anything further is required. Many thanks for your kind review.

Round 2

Reviewer 1 Report

Comments and Suggestions for Authors

The article is now ready for publication

Author Response

Dear Reviewer 1

Thank you for your response. I note no further changes are required. 

With best wishes

Caroline Laker

Reviewer 2 Report

Comments and Suggestions for Authors

I thank the Authors for the revision done and also for clarifying the qualitative and explorative nature of their study. Therefore, I have only one last suggestion to make, that is add this specification to the title, e.g.: "[....]: a qualitative preliminary study" or "[...]: preliminary results from a qualitative study".

Thank you, I send my best regards.

Author Response

Dear Reviewer

Thank you for your feedback. You suggestion has been included as follows:

The use of 360-degree video to reduce anxiety and increase confidence in mental health nursing students: a qualitative pre-liminary study. 

With best wishes,

Caroline Laker